# A study of roflumilast treatment on functional and structural changes in hippocampus in depressed Adult male Wistar rats

Ghida Hassan[1]*, Sherif A. Kamar[2,3], Hagar Yousry Rady[2,4], Dina Sayed Abdelrahim[5,6], Nesma Hussein Abdel Hay Ibrahim[7], Noha N. Lasheen[1,8]

**1** Medical Physiology Department, Faculty of Medicine, Ain Shams University, Cairo, Egypt, **2** Anatomy Department, Faculty of Medicine, Ain Shams University, Cairo, Egypt, **3** Faculty of Dentistry, Al-Ahliyya Amman University, Amman, Jordan, **4** Anatomy Department, Armed Forces College of Medicine, Cairo, Egypt, **5** Clinical Pharmacology department, Faculty of Medicine, Ain Shams University, Cairo, Egypt, **6** Pharmacology Department, Faculty of Medicine, Modern University for Technology and Information, Cairo, Egypt, **7** Medical Biochemistry and Molecular Biology Department, Faculty of Medicine, Ain Shams University, Cairo, Egypt, **8** Associate Professor of Physiology, Faculty of Medicine, Galala University, Suez, Egypt

* ghidamh.55@med.asu.edu.eg

**Data Availability Statement:** All relevant data are within the paper.

## Abstract

Depression is a common stress disability disorder that affects higher mental functions including emotion, cognition, and behavior. It may be mediated by inflammatory cytokines that interfere with neuroendocrine function, and synaptic plasticity. Therefore, reductions in inflammation might contribute to treatment response. The current study aims to evaluate the role of Protein Kinase (PKA)- cAMP response element-binding protein (CREB)- brain derived neurotropic factor (BDNF) signaling pathway in depression and the effects of roflumilast (PDE4 inhibitor) as potential antidepressant on the activity of the PKA-CREB-BDNF signaling pathway, histology, and pro-inflammatory cytokine production. Forty Adult male Wistar rats were divided into 4 groups: Control group, Positive Control group: similar to the controls but received Roflumilast (3 mg / kg / day) by oral gavage for the last 4 weeks of the experiment, Depressed group which were exposed to chronic stress for 6 weeks, and Roflumilast-treated group which were exposed to chronic stress for 6 weeks and treated by Roflumilast (3 mg / kg / day) by oral gavage for the last 4 weeks of the experiment. The depressed group showed significant increase in immobility time with significant decrease in swimming and struggling times, significant decrease in hippocampal PKA, CERB, BDNF, Dopamine, Cortisone, and Superoxide dismutase while hippocampal Phosphodiesterase-E4, Interleukin-6, and Malondialdhyde levels were significantly elevated. These findings were significantly reversed upon Roflumilast treatment. Therefore, it could be concluded that depression is a neurodegenerative inflammatory disease and oxidative stress plays a key role in depression. Roflumilast treatment attenuated the depression behavior in rats denoting its neuroprotective, and anti-inflammatory effects.

**Funding:** The authors received no specific funding for this work.

**Competing interests:** The authors have declared that no competing interests exist.

## Introduction

Depression is a widespread condition that affects people of all sexes, ages, socioeconomic classes, and species. According to the World Health Organization, depression is considered the main factor contributing to disability. In addition to feelings of guilt, poor focus, low self-esteem, sleep difficulties, and disturbed appetite, it also manifests as a lack of interest, desire, and pleasure [1].

Stress hormones are consistently over-secreted in depression, which leads to severe clinical manifestations and inhibits a sufficient response to stress. These hormones influence higher mental functions including emotion, cognition, and behavior in the brain by acting as neuromodulators. An important risk factor for making someone susceptible to stress-related diseases is chronic stress, which increases the production of corticosteroids and affects corticosteroid-receptor signaling [2]. According to a subsequent research, depression is a complex condition with many underlying causes and has been linked to an increased chance of developing serious medical conditions, including diabetes, cancer, Alzheimer's disease, epilepsy, cardiovascular disease, and stroke [1].

It was hypothesized that chronic stress might suppress neurogenesis, retract dendritic processes, and cause neuronal death in the hippocampus since research on people subjected to chronic stress revealed decreased hippocampal sizes [3].

Additionally, depression may cause disturbances in the way the brain regulates glucose and may be linked to cognitive decline. Leptin levels, negative mood, and sleep disruptions were revealed to be strongly positively correlated with one another [4]. Rats' ventral striatum's basal and feeding-stimulated dopamine (DA) release can both be reduced by leptin [5]. Furthermore, long-term inhibition of leptin signaling in the ventral tegmental area promotes locomotor activity whereas activation of leptin receptors suppresses firing of ventral tegmental area DA neurons [6]. Major depression may also result from an imbalance of ghrelin feedback in brain areas responsible for mood regulation. It has been discovered that ghrelin controls mood, dopaminergic characteristics, and the growth of the central nervous system. In both mice and people, it has antidepressant effects that have an impact on rewarding behavior [7].

Acute phase proteins, such as C-reactive protein and haptoglobin, as well as inflammatory cytokines like interleukin-1 (IL-1), interleukin-6 (IL-6), and TNF-alpha were shown to be typically related with illness behavior and depression, respectively. Such inflammatory cytokines can interact with any mediator associated with depression, including synaptic plasticity, neurotransmitter metabolism, and neuroendocrine function. Accordingly, inflammatory alterations may play a role in the development of depressive disorders, and as a result, decreasing inflammation may improve treatment outcomes [8].

It has been established that brain monoamines including norepinephrine (NE), serotonin (5-HT), and dopamine (DA), as well as their metabolites, play a role in the etiology of depression. Therefore, NE, 5-HT, and DA neurotransmitter deficits aggravate depressive diseases [9].

Numerous tools were examined, such as the Protein Kinase CAMP-Initiated Reactant Subunit Alpha (PRKACA) overflow, a significant cell signaling pathway that may be linked to the onset of depression by regulating synaptic pliancy, cytokinesis, transcriptional guidance, and the hypothalamic-pituitary-adrenal (HPA) axis [10]. Additionally, CREB is a transcriptional regulator that controls the expression of brain derived neurotropic factor (BDNF), a significant biomarker of depression [11], which is crucial for neuronal survival and function [12]. Recently, it was discovered that CREB/BDNF expression is essential for the pathophysiology of depression [13].

The chronic restraint stress (CRS) model is a well-established model for producing depressive-like behavior in animals [14]. It's interesting to note that CRS results in long-lasting

neuroinflammation, which is a key pathophysiologic cause of severe depressive disorders [15]. Furthermore, it was discovered that CRS reduced BDNF levels [16]. However, in the hippocampus and cortex of mammals, phosphodiesterase 4 (PDE4) is produced and selectively hydrolyzes cAMP to 5′-AMP. In recent years, it has been speculated as a possible depression drug target [17]. PDE4 inhibitors have been reported to reduce symptoms similar to depression. The PDE4 inhibitors that have currently been discovered include rolipram, roflumilast, ibudilast, MK-0952, HT-0712, GEBR-32a, and FFPM. It has been established that roflumilast plays a part in both synaptic plasticity and learning memory [18]. Roflumilast was discovered to enhance synaptic plasticity and memory formation in rats [19] and to lessen depression-like behavior in mice with Alzheimer disease [20]. According to these studies, roflumilast may one day be used as an antidepressant.

This study investigated the effects of roflumilast (PDE4 inhibitor) on the activity of the PRKACA-CREB-BDNF signaling pathway, pro-inflammatory cytokine production and change in hippocampal histology in male Wistar rats exposed to CRS model of depression. This study demonstrated the PKA-CREB-BDNF signaling pathway role in controlling stress-induced microenvironmental changes connected to a variety of neurological disorders, including depression.

## Materials and methods

### Ethics approval

All animal protocols complied with the Guide for the Care and Use of Laboratory Animals published by the US National Institutes of Health (NIH Publication No. 85–23, revised 2011) and were conducted with the approval of Physiology Department and by Faculty of Medicine-Ain Shams University research ethics committee, the approval number is **FMASU R 133/ 2022**.

### Animals

This study was performed on Forty adult Wistar male rats weighing between 150 and 200 g, purchased from the Holding Company for Biological Products and Vaccines (VACCERA), Helwan, Egypt. In the animal house of the Department of Clinical Pharmacology, Faculty of Medicine, Ain Shams University, rats were housed in groups of 4–5 rats in plexiglass cages with a mesh wire cover, with pelleted rat chow (Meladco, El Obour, Egypt) and tap water available ad libitum. To lessen the possibility of animal distress, rats were allowed to acclimatize to the laboratory conditions before any experiments for 2 weeks. Temperature was maintained at 25 °C, relative humidity 50–60%, and 12 h light/dark cycle at 5:00 a.m.–5:00 p.m. All efforts were made to reduce unneeded animal suffering or stress, and the handling was done by a trained laboratory specialist with the utmost care and cleanliness. Rats were chosen because their brains are relatively large, the animals are less timid and more intelligent, and some of their brain structure resembles the more primitive elements of human brains, and hence they can be used to model some human behaviors [21].

The methods of Euthanasia, at the end of the experiment animals were killed by overdose of anesthesia. Disposal of animal remains done by incineration.

Rats were randomly divided into four groups:

1. **Control group (n = 10)**; which were fed balanced chow diet-ad libitum-for 6 weeks.

2. **Positive control group (n = 10)**; which were similar to the controls but received Roflumilast (Western Pharmaceutical Industries) (WESTABREATH 500 ®) 3 mg / kg / day, by oral gavage, dissolved in distilled water for the last 4 weeks of the experiment [22].

3. **Depressed group (n = 10)**, which were exposed to chronic restraint stress for 6 weeks. The stress procedure was carried out in a different room by using a mild stress, restraint, which consisted of placing rats in Plexiglas tubes (25 x 8 cm) wide enough to allow comfortable breathing but restricting their movement, for 4 hours per day for 6 weeks performed at fixed time. Every stress session was carried out between 9:00 AM and 1:00 PM [23].

4. **Roflumilast-treated group (n = 10)**: rats were exposed to chronic restraint stress 4 hours per day for 6 weeks and were treated by Roflumilast (Western Pharmaceutical Industries) (WESTABREATH 500 [R]) 3 mg / kg / day, by oral gavage, dissolved in distilled water for the last 4 weeks of the experiment [22].

## Behavioral tests

At the end of 6 weeks, the experimental period, all animals were subjected to behavioral tests including Open Field Test (OFT) and Forced Swimming Test (FST) [24].

### Open Field Test (OFT)

Rats were given one hour to get used to the test room before the test began. Each rat was left alone for five minutes in the middle of a quadrangular arena (60x60 cm, divided into 16 identical squares) that was well-lit. The following characteristics were determined: the number of squares traversed (with all four feet crossed), the amount of time spent in the central zone (central four squares), the frequency of entry, and the latency to leave the central zone. After each rat, the OFT arena was cleaned with 10% alcohol [24].

### Forced Swimming Test (FST)

Rats swam in water that was 35 cm deep during FST in a vertical Plexiglass cylinder that was 22 cm in diameter and 50 cm in height. The water was kept at a constant 25 ˚C temperature. The rats swam for 15 minutes on the first day of training. On the next day (test day), rats swam for five minutes. Calculations of the immobility, swimming, and struggling times were made. Each rat underwent the test just once. Following each animal's test, the water was replaced [24].

### Sacrifice and sample collection

On the day of sacrifice, overnight fasted rats were anaesthetized with intraperitoneal injection of Pentobarbitone (40 mg/kg body weight) with booster doses as needed. The dose was given according to the guideline of rodent anesthesia analgesia formulary-UBC animal care [25].

When the stage of surgical anesthesia (judged by loss of withdrawal reflexes) had been reached, the animal was placed on its back and fixed on the operating table. The rat skull was opened, and the brain was dissected and washed by saline. The hippocampus was reached from the medial side after division of the brain at the mid sagittal plane into right and left hemispheres, followed by removal of the whole brain stem and cerebellum [26].

The hippocampus from all studied groups were stored at -80˚C for subsequent biochemical studies:

1-**Reverse Transcription Polymerase Chain Reaction (RT-PCR) method for PDE-4 and phosphorylated CERB determination**

## RNA extraction

Total RNA was extracted from homogenized tissues of all different groups using Direct-zol RNA Miniprep Plus (Cat# R2072, ZYMO RESEARCH CORP. USA) and then quantity and quality were assessed by Beckman dual spectrophotometer (USA).

## Real time PCR

SuperScript IV One-Step RT-PCR kit (Cat# 12594100, Thermo Fisher Scientific, Waltham, MA USA) was utilized for reverse transcription of extracted RNA followed by PCR. 96-well plate StepOne instrument (Applied Biosystem, USA) was used in a thermal profile as follows: 10 minutes at 45 °C for reverse transcription, 2 minutes at 98 °C for RT inactivation and initial denaturation by 40 cycles of 10 seconds at 98°C, 10 seconds at 55 °C and 30 second at 72 °C for the amplification step. After the RT-PCR run the data were expressed in Cycle threshold (Ct) for the target genes and housekeeping gene. Normalization for variation in the expression of target genes; *PDE-4, and CREB* were performed referring to the mean critical threshold (CT) expression values of *GAPDH* housekeeping gene by the ΔΔCt method. The relative quantitation (RQ) of each target gene is quantified according to the calculation of $2^{-\Delta\Delta Ct}$ method. Primers sequence for *PDE-4* gene was; forward 5′- GCTTGAACACCAACGTCCCACGGT -3′, and reverse 5′- GCTGAGGTTCTGGAAGATGTCGCAG -3′ (gene bank accession number NM_013101.4), *CREB* gene was; forward 5′- GAGACCTGGCCAGAGGATAC -3′, and reverse 5′- GTCAGTGAGCAAGAGAACGC -3′ (gene bank accession number NM_001005562.2), and *GAPDH* housekeeping gene was; forward 5′- CCTCGTCTCATAGACAAGATGGT -3′ and reverse 5′- GGGTAGAGTCATACTGGAACATG -3′ (gene bank accession number NM_001394060.2).

**2-Measurements of proteins by enzyme-linked immunosorbent assay (ELISA) in hippocampus tissues:**

## Preparation of tissue homogenate

Animals after they were sacrificed, tissues were washed thoroughly and rinsed with ice. They were gently blotted between the folds of a filter paper and weighed in an analytical balance. 10% of homogenate was prepared in 0.05 M phosphate buffer (pH 7) using a polytron homogenizer at 4°C. The homogenate was centrifuged at 10,000 rpm for 20 min for removing the cell debris, unbroken cells, nuclei, erythrocytes and mitochondria. The supernatant (cytoplasmic extract) was used to measure PRKACA, BDNF, IL-6, Dopamine, Cortisol, SOD and MDA according to the manual instructions [27].

## Determination of tissue protein

Protein content in the tissue was determined using Genei, Bangalore, protein estimation kit (Bangalore, Indian, Catalog # 2624800021730) [28].

All ELISA kits were measured by ELISA reader. Color absorbance was read at OD range 490 to 630 nm using an Enzyme-Linked Immuno-Sorbent Assay (ELISA) plate reader (Stat Fax 2200, Awareness Technologies, Florida, USA).

Protein Kinase CAMP-Activated Catalytic Subunit Alpha (PRKACA) was measured using Rat Protein Kinase A ELISA Kit (Bio-assay technology laboratory), the concentration of BDNF, IL-6 in tissues (pg/mg) were measured using Enzyme-linked Immunosorbent Assay Kit for BDNF and IL- 6 (Cloud Cone Corporation). DA and Corticosterone (CORT) were

measured using Mouse / Rat Dopamine ELISA Assay Kit (Eagle Bioscience, Catalog No, DOU39-K01) and Rat cortisol ELISA Kit (CUSABIO, Catalog No, CSB-E05112r) respectively.

## Measurement of oxidative stress markers

The activities of SOD (U/mg) & MDA (nmol/mg) were determined by using Superoxide Dismutase (SOD) Activity Assay Kit (BioVision, USA, Catalog #K335), and Lipid Peroxidation (MDA) Colorimetric/Fluorometric (BioVision, USA, Catalog # K739) respectively.

## (II) Histological studies

**Histological studies.**  The brain hemispheres were processed for paraffin blocks. Serial parasagittal sections were done. The sections were stained with hematoxylin and eosin [29]. Other specimens 1mm$^3$ were immediately fixed in 4% glutaraldehyde and processed for semi-thin sections. Sections were stained with Toluidine blue [30]. The sections were examined with an Olympus light microscope (CX31) equipped with digital camera in Anatomy department and were photographed.

**GFAP immunohistochemistry** was done to demonstrate the astrocytes. Sections were Deparaffinated, hydrated and incubated in blocking solution TBT [Tris Base Saline (TBS), 0.5 M pH 7.4 containing 3% (w/v) bovine serum albumin and 0.05% (v/v) Triton X-100]. Sections were incubated overnight at 4 ˚C in a humidified chamber with the anti- glial fibrillary acidic protein (GFAP) mouse monoclonal antibody at a 1:50 dilution. Slides were washed for 5 min in TBS. Immunodetection was performed using biotinylated anti-mouse immunoglobulins then incubation with 3,3 diaminobenzidine (DAB) chromogen (DAKO) in hydrogen peroxide for 5–10 minutes (brown staining). The sections were lightly counterstained with Mayer's hematoxylin, dehydrated and mounted [31].

## Statistical analysis

All results in this study were expressed as mean ± SEM. Statistical Package for the Social Sciences (SPSS, Inc., Chicago, IL, USA) program, version 20.0 was used to compare significance between groups. One Way ANOVA (Analysis Of Variance) for difference between means of different groups was performed on the results obtained in the present study. Differences were considered significant when P is ≤ 0.05.

## Results

### Behavioral tests, FST

As regards FST, immobility time was significantly elevated, meanwhile, swimming and struggle times were significantly lowered in untreated depressed groups compared to the controls (P≤0.001). However, immobility time was significantly lowered, swimming time was significantly elevated in treated depressed group compared to the untreated depressed rats (P≤0.001). Immobility and struggle times were significantly elevated in positive control rats, while swimming time was significantly lowered when compared to the control group (P≤0.001), as shown in Fig 1A.

### Behavioral tests, OFT

As regards OFT, total number of crossed squares was significantly decreased, meanwhile, time spend in the central zone and latency to leave the central zone were significantly increased in untreated depressed groups compared to the controls (P≤0.001, P≤0.05, P≤0.001 respectively). However, total number of crossed squares was significantly elevated, latency to leave

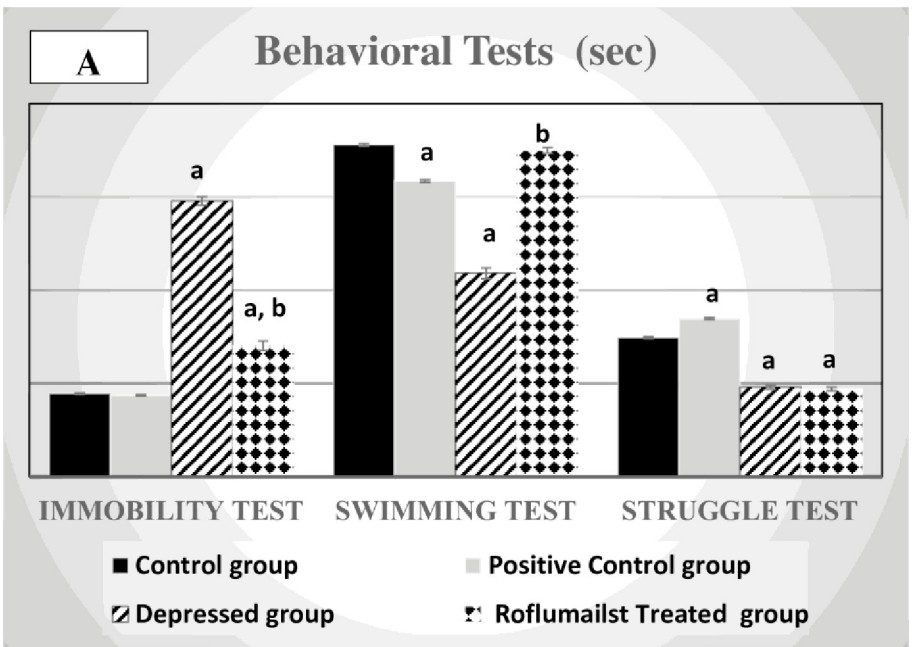

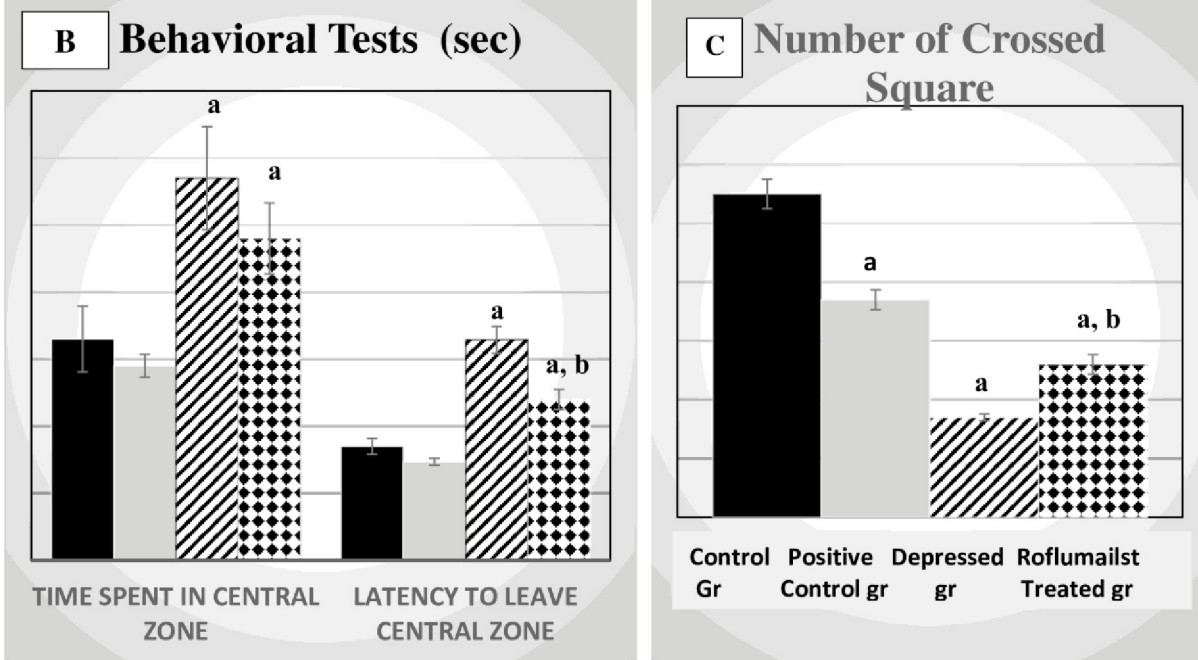

**Fig 1. Effects of Roflumilast on: (A) Forced swimming test and (B) Open Field test and (C) Number of crossed squares in Depressed Adult Male Wistar Rats (n = 10).** a: P<0.05 versus control group, b: P<0,05 versus depressed group.

the central zone was significantly lowered in treated depressed group compared to the untreated depressed rats (P≤0.001, P≤0.01 respectively). The total number of crossed squares and the latency to leave the central zone were significantly lowered in positive control rats, when compared to the control group (P≤0.001, P≤0.05 respectively), as shown in Fig 1B & 1C.

### Changes in PKA -CREB-BDNF signaling pathway

Regarding hippocampal PKA, CERB and BDNF levels were significantly lowered in untreated depressed group compared to the control group (P≤0.001 for each). they were significantly elevated in Roflumilast-treated depressed group compared to the untreated depressed rats (P≤0.001 for each). Non-significant changes were present in positive controls when compared to negative control ones, as shown in Fig 2B–2D.

### Changes in hippocampal PDE4 gene expression and IL-6

Meanwhile, hippocampal PDE4 and IL-6 levels, they were significantly elevated in untreated and Roflumilast-treated depressed groups compared to the controls (P≤0.001, P≤0.05, P≤0.001, P≤0.05 respectively). However, they were significantly lowered in Roflumilast-treated depressed group compared to the untreated depressed ones (P≤0.001 for both). No significant differences were observed in positive controls when compared to negative control rats, as shown in Figs 2A and 3A.

### Changes in DA and CORT

Both hippocampal DA and CORT were significantly decreased in untreated depressed group compared to the controls (P≤0.001 for each), meanwhile, both were significantly increased in Roflumilast-treated depressed group compared to the untreated depressed rats (P≤0.001 for each), as shown in Fig 3B & 3C.

### Oxidative stress markers changes

As shown in Fig 4, the untreated depressed group had significant higher hippocampal MDA level and significant lower hippocampal SOD level when compared to control ones (P≤0.001 for each), while Roflumilast-treated depressed rats had significantly lowered hippocampal MDA and significantly elevated hippocampal SOD compared to the untreated depressed rats (P≤0.001, for each). Non-significant differences were observed when comparing positive controls and negative control rats.

### Histological results

Histological examination of both the control and positive control groups showed almost the same results in all the examined stained sections, so the selected photomicrographs of the control group will be also representing for the positive control group.

Histological examination of both the control and positive control groups showed almost the same results in all the examined stained sections.

### H&E stained sections

**Examination of the H&E stained sections of CA1 region of the Control group** showed that the hippocampal formation was formed of two regions; the hippocampus proper (Cornu Ammonis, with its four regions; CA1, CA2, CA3 and CA4) and the dentate gyrus (DG) (Fig 5). The cells forming the CA1 region were arranged in three layers: the polymorphic layer, the pyramidal cell layer and the molecular layer. The pyramidal cell layer appeared to be the principal layer of the hippocampus proper and was formed of pyramidal cells with large rounded vesicular nuclei and prominent nucleoli. Both the polymorphic and the molecular layers showed scanty scattered glial cells (Fig 6a).

**Examination of the H&E stained sections of CA1 region of the Positive control group** showed that the pyramidal cell layer was formed of pyramidal cells with large rounded

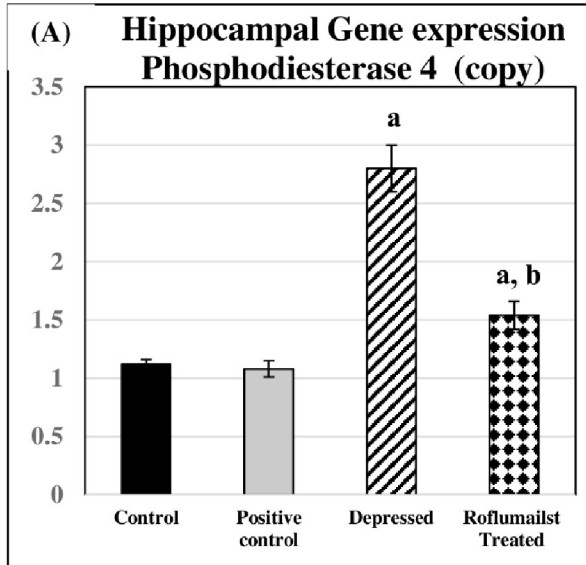

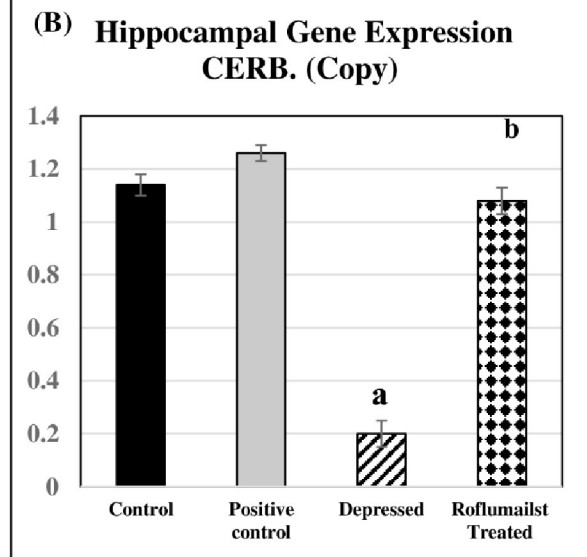

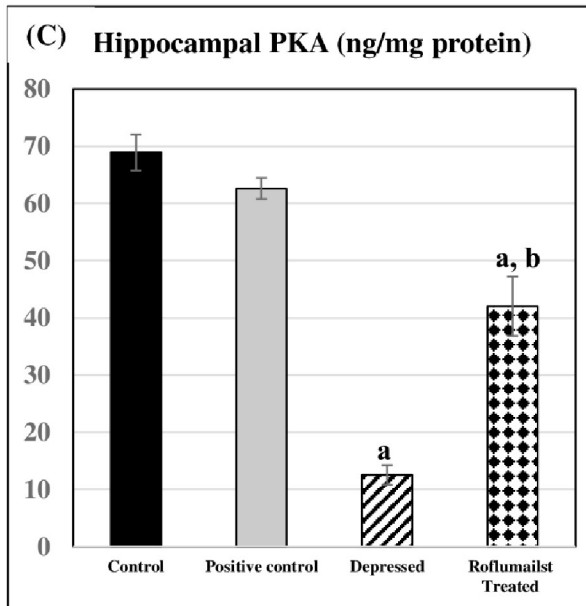

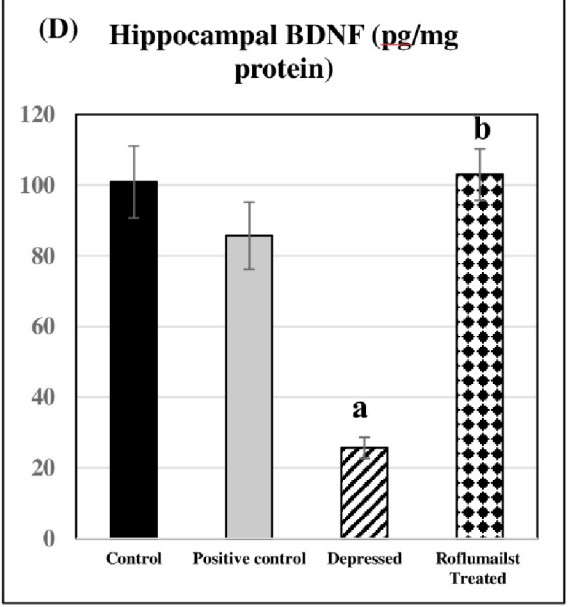

**Fig 2. Effects of Roflumilast on (A): Phosphodiesterase 4, gene expression (B): CERB gene expression, (C): level of PKA and (D): level of BDNF in Depressed Adult Male Wistar Rats (n = 10).** a: $P < 0.05$ versus control group, b: $P < 0.05$ versus depressed group.

vesicular nuclei and prominent nucleoli. Both the polymorphic and the molecular layers showed scanty scattered glial cells (Fig 6b).

**Examination of the H&E stained sections of CA1 region of the Depressed group** revealed that most of the pyramidal cells had irregular outlines with dark deeply stained nuclei with pericellular spaces (vacuolations) around some of the pyramidal cells. Also, pericellular vacuolation around the glial cells and spacing within the molecular layer and the polymorphic layers were detected (Fig 6c).

**Examination of the H&E stained sections of CA1 region of the Roflumilast-treated group** showed that most of the pyramidal cells had large rounded vesicular nuclei with

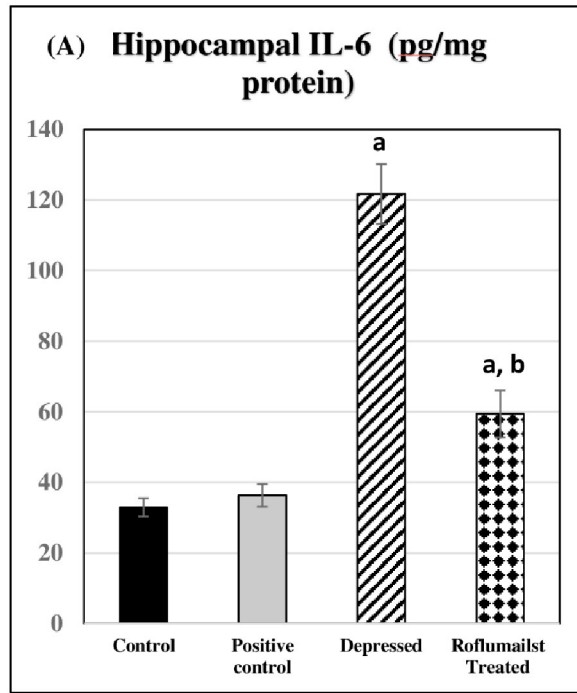

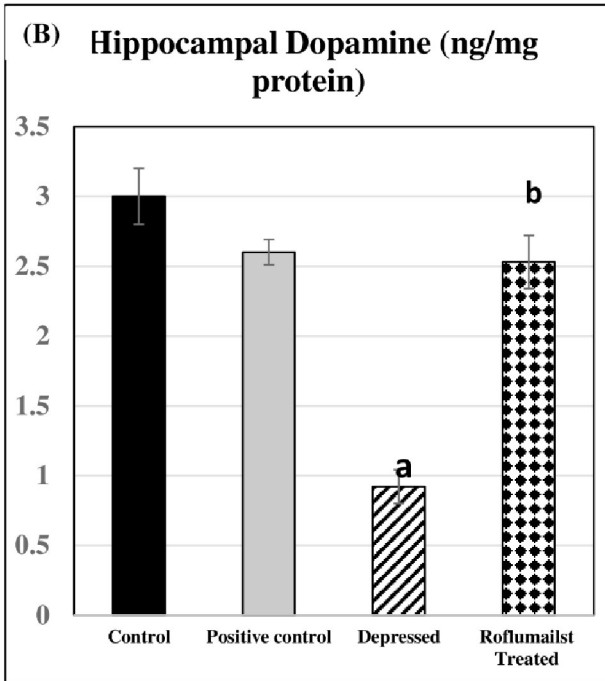

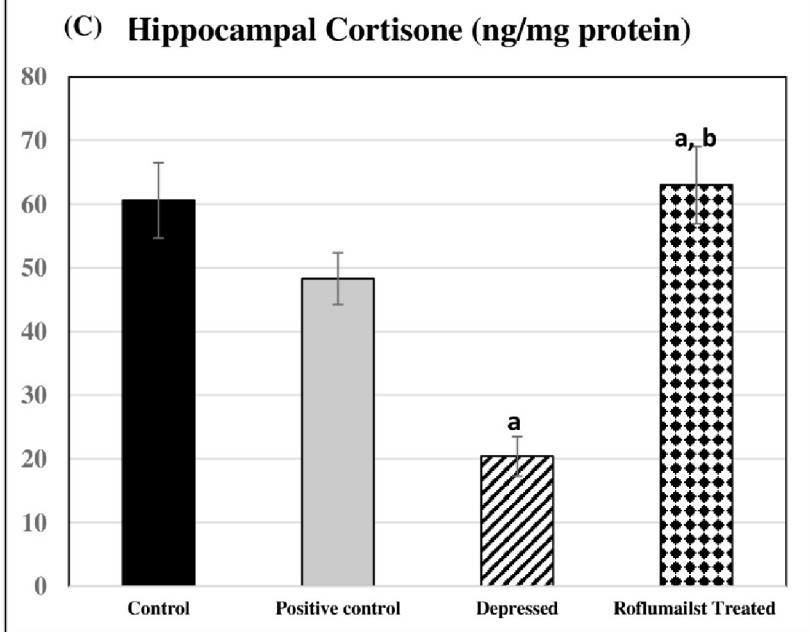

**Fig 3. Effects of Roflumilast on hippocampal levels of: (A) IL6, (B) dopamine and (C) cortisone in Depressed Adult Male Wistar Rats (n = 10)a: P<0.05 versus control group, b: P<0.05 versus depressed group.**

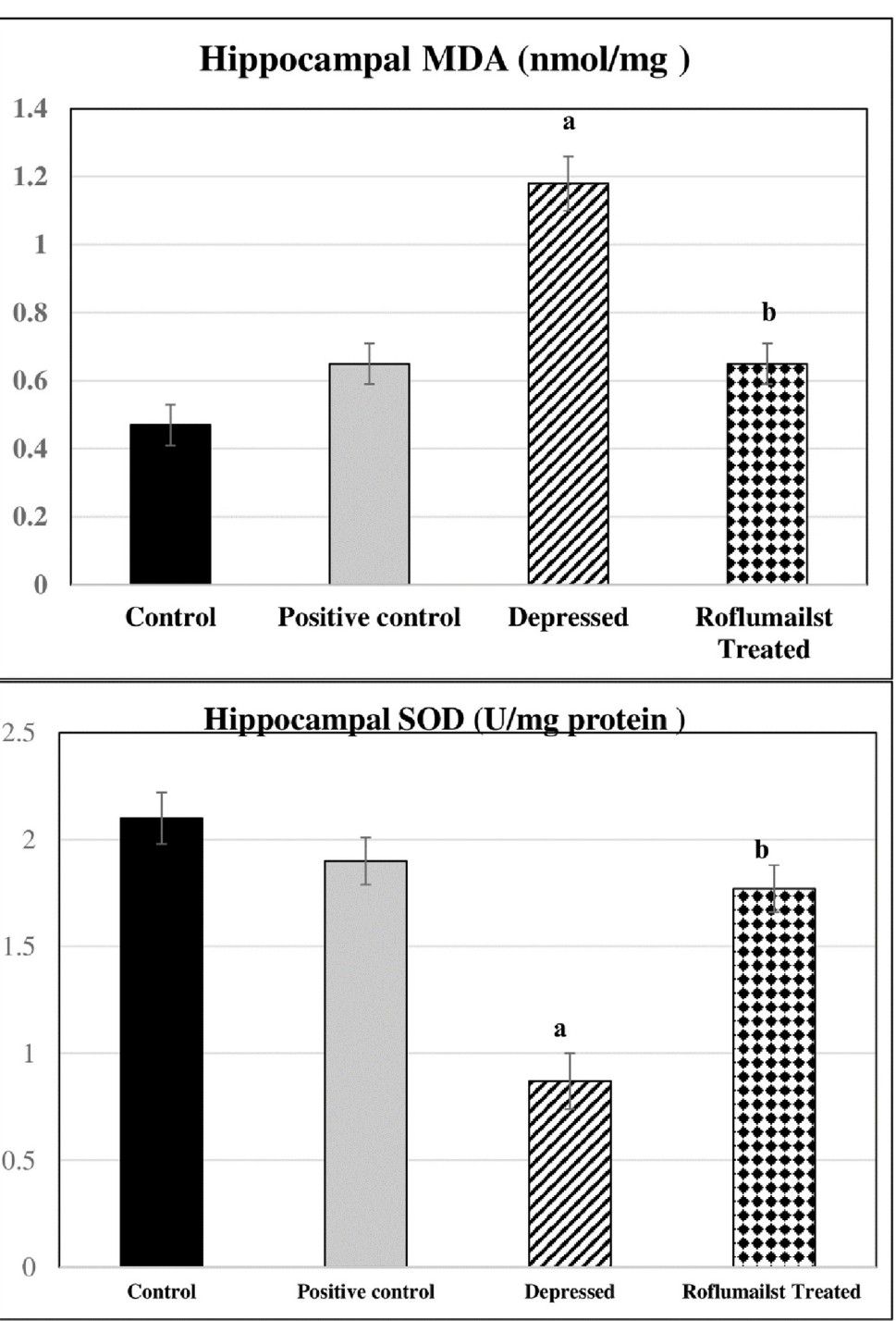

**Fig 4. Effects of Roflumilast on hippocampal oxidative stress markers (Malondialdhyde "MDA" and Superoxide Dismutase "SOD") in depressed Adult male Wistar rats (n = 10).** a: $P<0.05$ versus control group, b: $P<0.05$ versus depressed group.

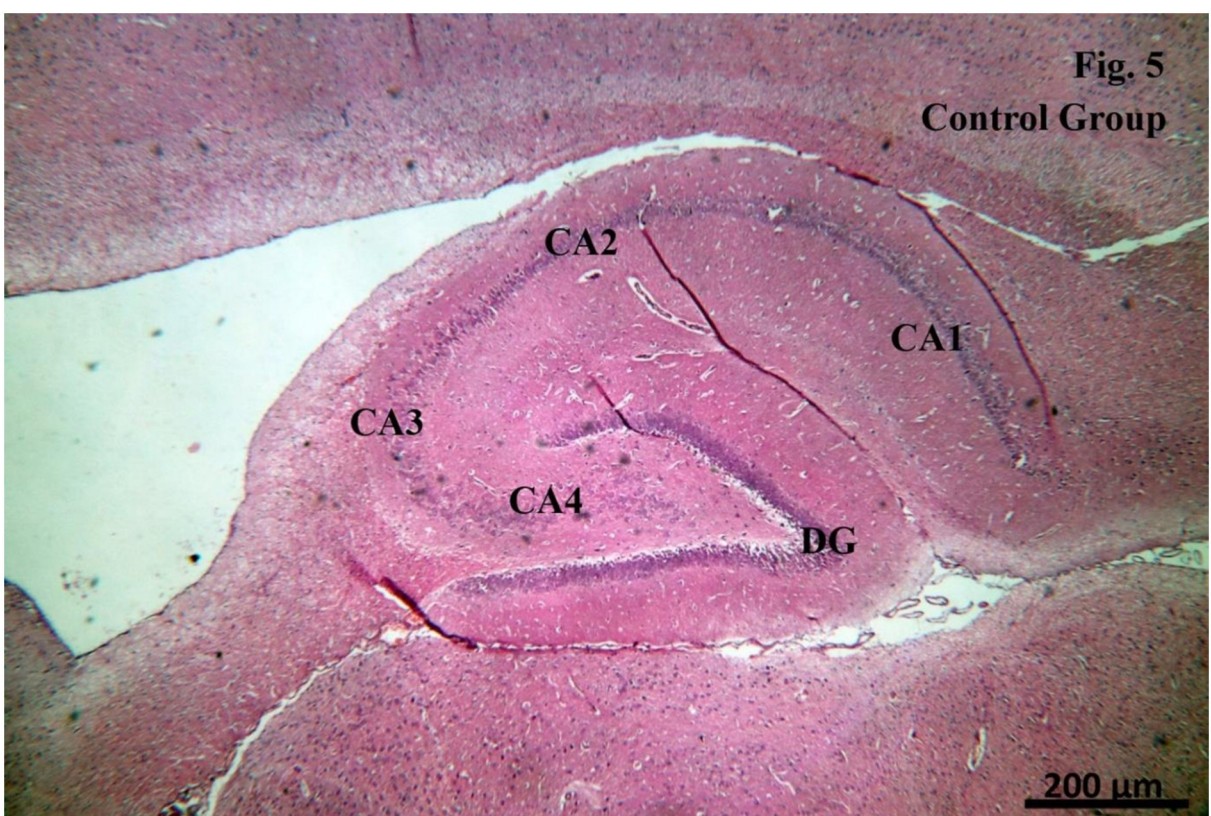

**Fig 5. Examination of the H&E stained sections of CA1 region of the Control group.**

prominent nucleoli. Few pyramidal cells showed irregular outlines and variable sizes with darkly stained nuclei and pericellular spaces. Pericellular vacuolation around the glial cells and spacing within the molecular layer and the polymorphic layers were detected (Fig 6d).

## Toliudine blue stained semithin sections

**Examination of the toliudine blue stained semithin sections of CA1 region of the Control group** showed that the pyramidal layer was formed of densely packed pyramidal cells with large rounded vesicular nuclei and prominent nucleoli with projecting processes. The presence of glial cells with pale ovoid nuclei and lightly stained cytoplasm was detected within the molecular and polymorphic layers (Fig 7a).

Examination of the toliudine blue stained semithin sections of CA1 region of the Positive control group showed that the pyramidal layer was formed of densely packed pyramidal cells with large rounded vesicular nuclei and prominent nucleoli (Fig 7b).

Examination of the toliudine blue stained semithin sections of CA1 region of the Depressed group showed the pyramidal cells with irregular outlines, variable sizes and with dark deeply stained nuclei. Many glial cells with pale ovoid nuclei and lightly stained cytoplasm throughout the layers were also detected (Fig 7c).

Examination of the toliudine blue stained semithin sections of CA1 region of the Roflumilast-treated group revealed that most of the pyramidal cells within the pyramidal layer appeared densely packed with large rounded vesicular nuclei, prominent nucleoli and with projecting processes. Very few pyramidal cells had dark deeply stained nuclei with irregular

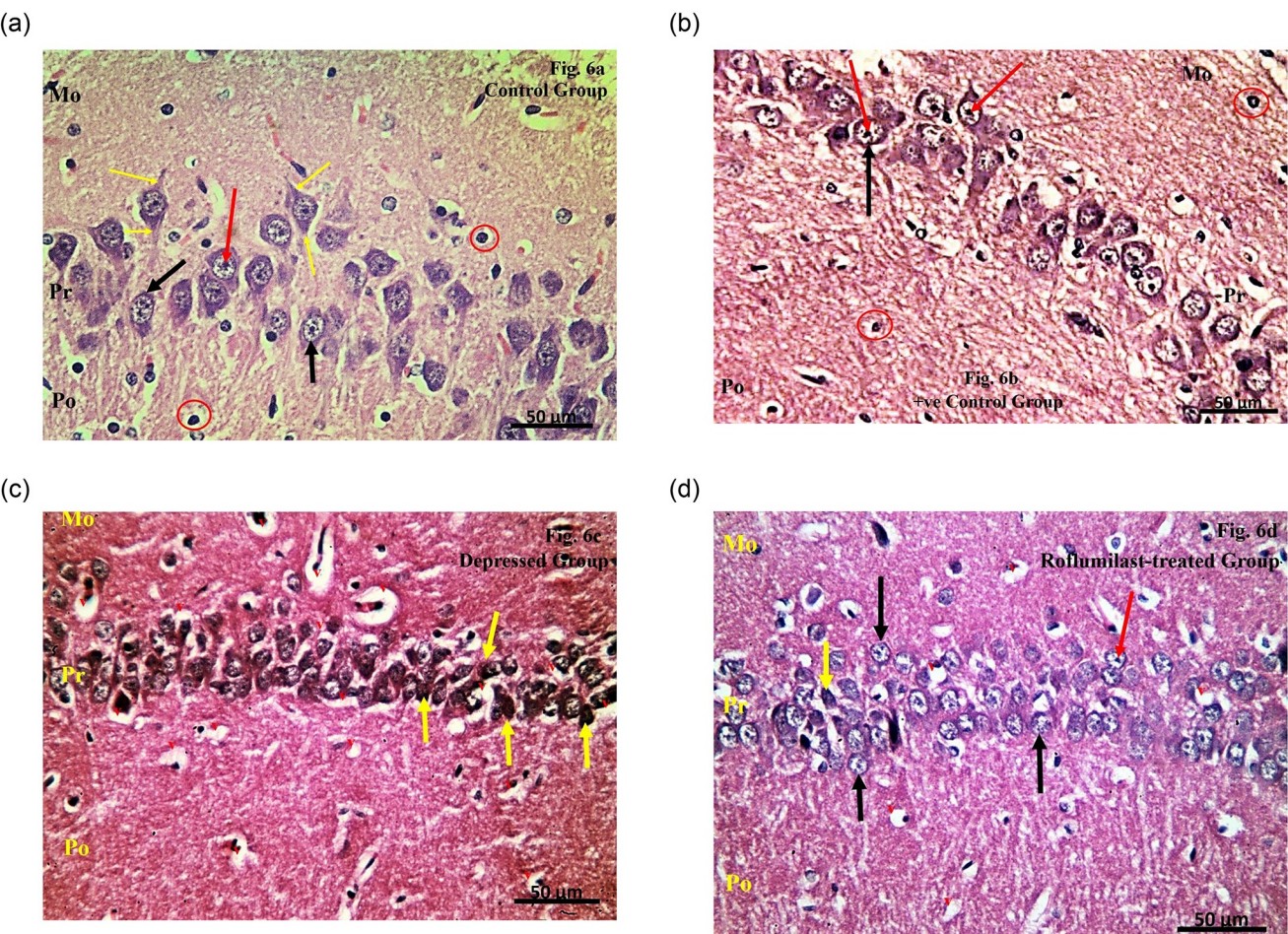

**Fig 6.** **a**. Examination of the H&E stained sections of CA1 region of the Control group. **b**. Examination of the H&E stained sections of CA1 region of the Positive control group. **c**. Examination of the H&E stained sections of CA1 region of the Depressed group. **d**. Examination of the H&E stained sections of CA1 region of the Roflumilast-treated group.

outline. Few glial cells with pale ovoid nuclei and lightly stained cytoplasm could be seen within the within the molecular and polymorphic layers (Fig 7d).

## Immunohistochemical (GFAP) stained sections

**Examination of the GFAP stained sections of CA1 region of the Control group** showed apparent minimal GFAP immunostaining of the astrocytes represented by minimal brownish color (Fig 8a).

**Examination of the GFAP stained sections of CA1 region of the Positive control group** showed apparent minimal GFAP immunostaining of the astrocytes represented by minimal brownish color (Fig 8b).

**Examination of the GFAP stained sections of CA1 region of the Depressed group** showed apparent extensive GFAP immunostaining of the astrocytes with ramifying processes along the layers (Fig 8c).

**Examination of the GFAP stained sections of CA1 region of the Roflumilast-treated group** revealed apparent minimal GFAP immunostaining of the astrocytes (Fig 8d).

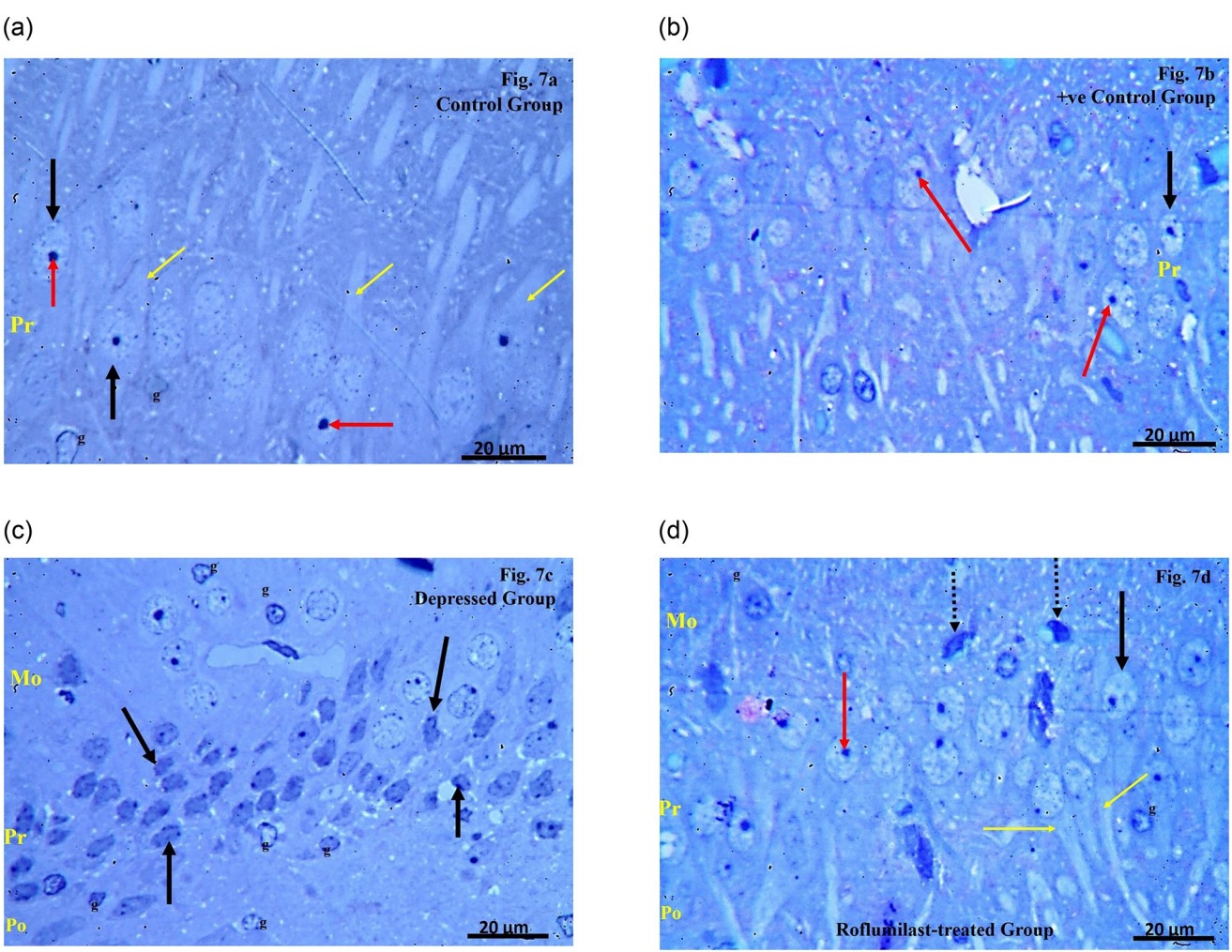

**Fig 7. a**. Examination of the toliudine blue stained semithin sections of CA1 region of the Control group. **b**. Examination of the toliudine blue stained semithin sections of CA1 region of the Positive control group. **c**. Examination of the toliudine blue stained semithin sections of CA1 region of the Depressed group. **d**. Examination of the toliudine blue stained semithin sections of CA1 region of the Roflumilast-treated group.

## Correlation studies

In the current study, significant negative correlations exist in all studied groups between malondialdhyde (MDA) and cAMP response element-binding protein (CERB), Protein Kinase CAMP-Activated Catalytic Subunit Alpha (PRKACA), Brain derived neurotropic factor (BDNF), Dopamine (DA) and Superoxide Dismutase (SOD) while a significant positive correlation exists between MDA and Phosphodiesterase-4 (PDE-4), Interleukin-6 (IL-6) and Corticosterone (CORT) (Table 1). Also, significant negative correlations were observed between PDE4 and CERB, PRKACA, BDNF while there was a significant positive correlation exists between PDE4 and IL-6 (Table 2). In addition, there were significant negative correlations between DA and the immobility time, time Spent in Central Zone, Latency to Leave Central Zone, while significant positive correlations were found between DA and swimming time, struggle time and total no. of crossed squares, as shown in Table 3.

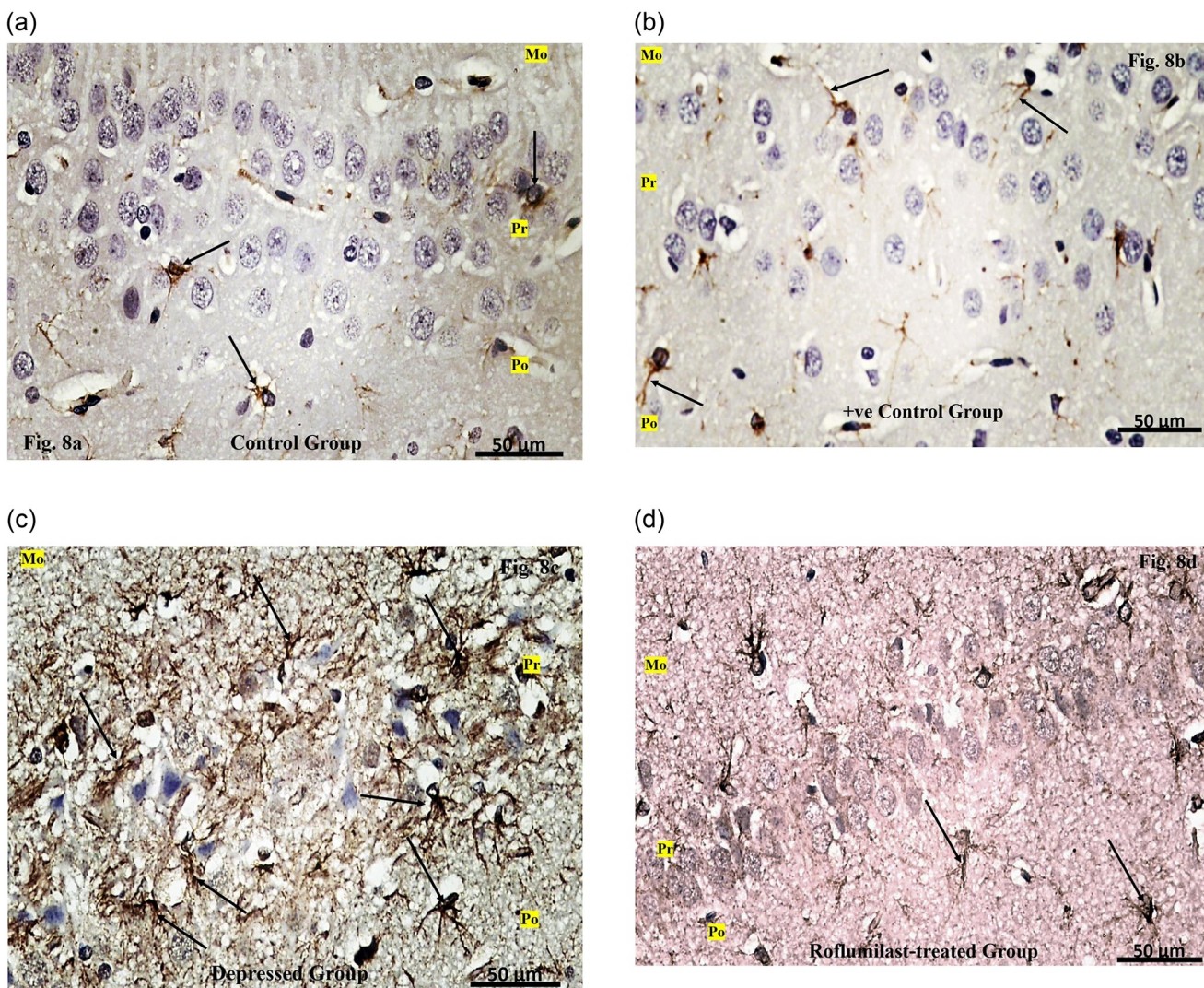

**Fig 8.** **a**. Examination of the GFAP stained sections of CA1 region of the Control group. **b**. Examination of the GFAP stained sections of CA1 of the Positive control group. **c**. Examination of the GFAP stained sections of CA1 region of the Depressed group. **d**. Examination of the GFAP stained sections of CA1 region of the Roflumilast-treated group.

## Discussion

Due to the recent trend toward identifying novel therapeutic targets for depression, this study investigated the effects of chronic restraint stress (CRS)-induced depression on many neurotransmitters and mediators in hippocampi, also treatment with roflumilast and its role in alleviation of such changes. This study revealed CRS-induced depressive-like behavior manifested

**Table 1. Correlations of malondialdhyde (MDA) and other parameters in all studied groups: (cAMP response element-binding protein (CERB), Protein Kinase CAMP-Activated Catalytic Subunit Alpha (PRKACA), Brain derived neurotropic factor (BDNF), Dopamine (DA), Superoxide Dismutase (SOD) Phosphodiesterase-4 (PDE-4), Interleukin-6 (IL-6) and Corticosterone (CORT)).**

|   | CERB | PRKACA | BDNF | DA | SOD | PDE4 | IL-6 | CORT |
|---|---|---|---|---|---|---|---|---|
| **R** | -0.65 | -0.72 | -0.69 | -0.67 | -0.72 | 0.66 | 0.78 | 0.64 |
| **P** | $\leq 0.01$ | $\leq 0.001$ | $\leq 0.001$ | $\leq 0.001$ | $\leq 0.001$ | $\leq 0.001$ | $\leq 0.001$ | $\leq 0.01$ |

**Table 2. Correlations of Phosphodiesterase-4 (PDE-4) and cAMP response element-binding protein (CERB), Protein Kinase CAMP-Activated Catalytic Subunit Alpha (PRKACA), Brain derived neurotropic factor (BDNF) and Interleukin-6 (IL-6).**

|  | CERB | PRKACA | BDNF | IL-6 |
|---|---|---|---|---|
| R | -.0871 | -0.894 | -0.761 | 0.809 |
| P | $\leq 0.001$ | $\leq 0.001$ | $\leq 0.001$ | $\leq 0.001$ |

by higher FST immobility time with reduced swimming and struggling times, denoting the development of anhedonia and behavioral despair, key features of human depressive symptomatology [32]. The decreased number of crossed squares and the increased latency to leave central zone in open field test (OFT) reflect a freezing behavior, an index of the depression-associated anxiety [33]. The depressive-like behavior was associated with alteration in some biochemical parameters such Protein Kinase cAMP-Activated Catalytic Subunit Alpha (PRKACA), Phosphorylated cAMP response element-binding protein (CREB) and Brain derived neurotrophic factor (BDNF), Phosphodiesterase E4 (PDE4).

Meanwhile, roflumilast treatment reversed CRS-depressive-like behavior. Similarly, there was lowered immobility time in APP/PS1 transgenic mice, a widely used model of Alzheimer's disease (AD) Wang et al. (2020) [20]. In line, roflumilast reversed cognition deficits and depression-like effects in APP/PS1 transgenic mice [34]. Roflumilast enhanced imipramine's effect, suggesting its superior efficacy as an adjuvant to standard antidepressants [35]. Our finding is aligned with the study of Hasan et al. (2022) [36], which demonstrated that roflumilast increased rats 'exploratory potential as it increases number of crossed squares in OFT. In line, Waltrick et al. (2023) [37] demonstrated the antidepressive effects of roflumilast in the form of increased latency to leave central zone.

The underlying mechanism of depression was suggested to be mediated by chronic hypercortisolemia, which results in excessive release of reactive oxygen species (ROS), increased NF-B transcriptional activity, and decreased synthesis of neurotrophic factors in many glucocorticoid receptor rich brain areas, such as the prefrontal cortex, amygdala, and hippocampus, with subsequent induction of structural and functional deficits. This sustained "stress responsive" HPA axis hyperactivity has been linked to the pathophysiology of depression and its associated cognitive dysfunction [38, 39]. PDE inhibitors were, also, suggested to modulate brain functions by altering the cAMP/cGMP/BDNF signaling [40].

In the current study, depression was manifested by reduction in hippocampal Protein Kinase cAMP-Activated Catalytic Subunit Alpha (PRKACA), Phosphorylated cAMP response element-binding protein (CERB) and Brain derived neurotrophic factor (BDNF) levels, however, there was a rise of hippocampal Phosphodiesterase E4 (PDE4) level. These changes were reversed by treatment with roflumilast.

BDNF has been found to be essential in sustaining synaptic plasticity, synapse development, and neuronal proliferation. The phosphorylation of cAMP-response element-binding protein (CREB) regulates BDNF expression, which is a target for alleviating neuron injury. CREB phosphorylation is controlled by cAMP PKA [41]. PDE4, a cyclic adenosine monophosphate

**Table 3. Correlations of Dopamine (DA) and findings of behavioral tests in all studied parameters.**

|  | Immobility Time | Time Spent in Central Zone | Latency To Leave Central Zone | Swimming Time | Struggle Time | Total No. Of Crossed Squares |
|---|---|---|---|---|---|---|
| R | -0.885 | -0.485 | -0.664 | 0.844 | 0.456 | 0.741 |
| P | $\leq 0.001$ | $\leq 0.01$ | $\leq 0.001$ | $\leq 0.001$ | $\leq 0.05$ | $\leq 0.001$ |

(cAMP) hydrolase, uses the cAMP/PKA signaling pathway to control CREB phosphorylation [42].

In the current study, there was downregulation in hippocampal gene expression of BDNF, denoting reduced intracellular signaling pathways. The lowered BDNF expression confirmed by the significant decrease in CERB and PKA may explain the depressive behavior in depressed rats, herein. Moreover, PDE4 was significantly increased in depressed rats reflecting its role in inducing depression. Similarly, Zhong et al. (2019) [43], suggested that reduction of brain-derived neurotrophic factor (BDNF) causes neuronal apoptosis. In line, low BDNF levels may be lowered in the cerebral cortex of depressed and suicide subjects [44]. In support, there is a significant inverse relationship between BDNF blood levels and depression scores in depressed patients [45].

As a result, roflumilast, a selective inhibitor of phosphodiesterase-4 (PDE-4) that promotes BDNF expression and alleviates depression-induced neuronal injury and apoptosis, induces CREB phosphorylation, similar to Zhang et al. [46]. They suggested that roflumilast affected BDNF through acting on the cAMP/CREB signaling pathway, by inhibiting PDE-4 actions. The elevated IL-6 levels in untreated and Roflumilast-treated depressed groups, despite being lowered in Roflumilast-treated depressed group compared to the untreated depressed ones could be explained by chronic stress-induced activation of inflammatory pathways triggering the development of depression, thus depressive disorders could partly be explained by some inflammatory mediators, namely increased IL-6.

Also, the higher IL-6 levels in depressed rats, herein, could be attributed to depression-induced alteration of local brain activity [47]. As cytokines affect brain activity through neuro-transmitter-depletion, neuroendocrine changes, and neural plasticity pathways, inflammatory cytokines may inhibit neurogenesis. In support, anti-inflammatory administration caused recovery or enhancement of neurogenesis [48]. Thus, chronic stress was found to stimulate cytokine release to blood and microglia hindering neurogenesis. Depression was explained by the neurogenic theory which suggested that impaired neurogenesis caused depression, so, anti-depressive drugs could improve neurogenesis [49].

The IL-6-lowering effect of roflumilast, in the present study, could be due to its targeting effects in the neurotrophic pathway in ameliorating the depressive manifestations through lowering the neurotrophic factors, altering the plasticity and neurogenesis in the brain, thereby having anti-depressant potentials. Therefore, the reduced immobility time and the anti-depressant activity, observed herein, could be explained by adjusting neurotrophic pathways.

Therefore, depression, a neurodegenerative inflammatory disease, may be caused by oxidative stress. This was confirmed through the correlation study which showed that the increased oxidative markers in the depressed rats evidenced by higher MDA resulted in decline in the CERB, PRKACA and BDNF. The latter two factors are essential for the neuronal viability and synaptic plasticity in the adult brain that are important in cognitive flexibility and an increased ability to adapt with environmental challenges that may precipitate or exacerbate depressive episodes. On the other side, the increased MDA resulted in significant increase in the PDE4, IL-6.

The lowered hippocampal dopamine (DA), herein, in untreated depressed rats may be attributed to stress-induced downregulation of dopaminergic pathways causing depression, in accordance, many studies [50, 51]. The higher DA levels caused by Roflumilast administration, herein, may be attributed to an increase of DA synthesis and turnover in striatum, thereby the dopamine D1 receptor signaling is enhanced in frontal cortical pyramidal neurons [52].

Interestingly, hippocampal corticosteroid was significantly decreased in untreated depressed rats, despite being significantly elevated in treated depressed ones. This may be attributed to altered corticosteroid level caused by stress exposure in the current study.

Lowered blood corticosteroid concentrations were explained by increased metabolism and/or tissue uptake and being independent on steroidogenesis [53]. They suggested that the decline in circulating corticosterone in rats exposed to repeated chronic stressor may be a compensatory mechanism for the higher excitability of serotonergic neurons to limit the deleterious effects of repeated chronic stressor.

Moreover, stress-induced depressive-like behavior, herein, was associated by prominent oxidative stress denoted by higher hippocampal MDA level and lower hippocampal SOD, while roflumilast has antioxidant effects. Therefore, the structural derangement observed in hippocampi of depressed rats is explained by the prominent oxidative stress. Also, roflumilast reserved almost normal hippocampal structural features by its antioxidant defense systems, such as the enzymes superoxide dismutase and catalase [54].

Therefore, the altered hippocampal mediators could be attributed to the evident oxidative stress through raising the neuronal inflammatory agents and apoptotic mediators, leading to neuronal death. In addition, the inflammatory mediators may impair the glial cells, namely astrocytes and microglia, producing neurodegenerative changes [55], which may explain the higher hippocampal IL-6. Moreover, oxidative stress may be the causative agent of dopaminergic neurons damage [56], evidenced by the reduction in DA level in the current study. These findings are in line with Tuon et al. [57]. They suggested that neurodegenerative changes, elevated lipid peroxidation, and lowered superoxide dismutase activity may produce major depressive disorders.

Although corticosteroid level was decreased in the depressed group, there was a significant positive correlation between MDA and corticosteroid level that resulted in altered hippocampal function.

On the other hand, roflumilast treatment ameliorated the oxidative stress evidenced by significant decrease in MDA and significant increase in the SOD, in accordance with Bhatt et al. (2020) [56].

Therefore, the prolonged oxidative stress led to increased PDE4 which in turn decreased CERB, PRKACA and BDNF and increased IL-6, enhancing neuronal susceptibility to injury and development of depression. Besides, it impaired neurogenesis and synaptic plasticity.

In addition, the prominent oxidative stress resulted in lowered antioxidant SOD and reduced DA which is normally responsible for the motivation, arousal, reinforcement, and reward encouraging the brain cells to act in a pleasurable, excitable behaviors. This was confirmed by the correlation study that showed a decline in the motor activity in the depressed group such as swimming time, struggle time and total no. of crossed squares and the rise of the depressed behavior such as significant prolongation in the immobility time, time spent in central zone, latency to leave central zone.

Regarding the histopathological changes in the current study, the PV1 interneuron numbers in hippocampal subdivision were reduced in long-term stress similar to many light microscopic studies [58–60]. The Hilar region including PV1 neuronal degenerative changes and/or necrosis were found in cases of chronic stress [61]. Similarly, the pericellular vacuolation around the glial cells and spacing within the molecular layer and the polymorphic layers were detected, indicating degenerative changes. Meanwhile roflumilast-treated group had almost normal hippocampal neuronal appearance.

Selective GFAP immunohistochemical reaction was used to assess the numerical and structural integrity of hippocampal astrocytes according to Webster et al. [62]. In the current study an apparent extensive GFAP reaction of the astrocytes with ramification of their processes along the layers of CA1 in depressed rats, which disagree with Ayuob et al. [63], who found a decrease of labeling of astrocytes adjacent to blood vessels as a characteristic feature for schizophrenia and major depression.

On the other hand, astrocytic GFAP reaction was found to be minimal in CA1 region of the Roflumilast-treated group giving apparent similar picture to the control ones.

## Conclusion

Oxidative stress resulted in significant decrease in the CERB, PRKACA and BDNF and significant increase in the PDE4, IL-6. This led to decrease in the cognitive flexibility and decreased the ability to adapt with environmental challenges that may precipitate or exacerbate depressive episodes. In addition, structural affection was proved histomorphologically in different parts and layers of the hippocampus as a result of depression and was further proved by immunohistochemical reaction (GFAP).

All the previous findings were reversed by Roflumilast treatment (PDE4 inhibitor) which attenuated the depression behavior in rats denoting its neuroprotective, and anti-inflammatory effects.

## Limitations of the study

One of the strongest challenges that we faced in our study is the sample size. The hippocampus size was small to perform all the above mentioned biochemical and histological studies, although we extracted the hippocampus from both hemispheres. In addition, we faced difficulty in purchasing the required amount of the drug o be able to perform the whole study.

## Author Contributions

**Conceptualization:** Ghida Hassan, Sherif A. Kamar, Dina Sayed Abdelrahim, Noha N. Lasheen.

**Data curation:** Ghida Hassan.

**Formal analysis:** Ghida Hassan.

**Investigation:** Ghida Hassan, Sherif A. Kamar, Dina Sayed Abdelrahim, Nesma Hussein Abdel Hay Ibrahim.

**Methodology:** Ghida Hassan, Sherif A. Kamar, Hagar Yousry Rady, Dina Sayed Abdelrahim, Noha N. Lasheen.

**Project administration:** Ghida Hassan, Noha N. Lasheen.

**Resources:** Ghida Hassan, Sherif A. Kamar, Dina Sayed Abdelrahim, Nesma Hussein Abdel Hay Ibrahim.

**Supervision:** Ghida Hassan.

**Visualization:** Ghida Hassan, Noha N. Lasheen.

**Writing – original draft:** Ghida Hassan, Sherif A. Kamar, Hagar Yousry Rady, Dina Sayed Abdelrahim, Nesma Hussein Abdel Hay Ibrahim, Noha N. Lasheen.

**Writing – review & editing:** Ghida Hassan, Sherif A. Kamar, Hagar Yousry Rady, Dina Sayed Abdelrahim, Nesma Hussein Abdel Hay Ibrahim, Noha N. Lasheen.

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
