## [Decision Letter · Decision Letter 0]

7 Aug 2023

PONE-D-23-15624A Study of Roflumilast Treatment on Functional and Structural Changes in Hippocampus in Depressed Adult Male Wistar RatsPLOS ONE

Dear Dr. Hassan,

Thank you for submitting your manuscript to PLOS ONE. After careful consideration, we feel that it has merit but does not fully meet PLOS ONE’s publication criteria as it currently stands. Therefore, we invite you to submit a revised version of the manuscript that addresses the points raised during the review process.

This article discusses an interesting point concerning depression behavior which is a common stress disability disorder that affects higher mental functions including emotion, cognition, and behavior. Unfortunately, there are several major concerns raised by the reviewers in the attached documents that will preclude further consideration at this time. We feel that the work is significant and of interest to our readers, and therefore we are willing to consider a thoroughly revised manuscript addressing all of the points raised by the reviewers. Please submit your revised manuscript by Sep 21 2023 11:59PM. If you will need more time than this to complete your revisions, please reply to this message or contact the journal office at plosone@plos.org. Please include the following items when submitting your revised manuscript:A rebuttal letter that responds to each point raised by the academic editor and reviewer(s). You should upload this letter as a separate file labeled 'Response to Reviewers'.A marked-up copy of your manuscript that highlights changes made to the original version. You should upload this as a separate file labeled 'Revised Manuscript with Track Changes'.An unmarked version of your revised paper without tracked changes. You should upload this as a separate file labeled 'Manuscript'.

We look forward to receiving your revised manuscript.

Kind regards,

Abeer El Wakil, PhD

Academic Editor

PLOS ONE

https://www.sciencedirect.com/science/article/abs/pii/S0014299920305033?via%3Dihub

https://karger.com/cpb/article/31/6/761/71413/Molecular-Mechanisms-of-Depression-Perspectives-on

In your revision ensure you cite all your sources (including your own works), and quote or rephrase any duplicated text outside the methods section. Further consideration is dependent on these concerns being addressed.

Reviewers' comments:

Reviewer's Responses to Questions

**Comments to the Author**

1. Is the manuscript technically sound, and do the data support the conclusions?

Reviewer #1: Yes

Reviewer #2: Partly

2. Has the statistical analysis been performed appropriately and rigorously? 

Reviewer #1: Yes

Reviewer #2: Yes

3. Have the authors made all data underlying the findings in their manuscript fully available?

Reviewer #1: Yes

Reviewer #2: No

4. Is the manuscript presented in an intelligible fashion and written in standard English?

Reviewer #1: No

Reviewer #2: Yes

5. Review Comments to the Author

Reviewer #1: The idea of the article is interesting, and the manuscript is well written. The experiments are described in sufficient detail. However, a major revision is recommended. Please check the attached file.

Reviewer #2: Need to revise this manuscript before can be accepted.

1. In abstract, anti-apoptotic pathway suggest remove, cause not measured in this study.

2. The positive drug group suppose received drug usually used to treat depression.

3. Please explained from which hemisphere brain was extracted for different methods in this study.

4. Please explain why positive group have elevated immobility and struggle time compared to control group?

5. Please changes the group legend, between positive control and depressed groups. For OFT there was not total number of crossed squares data?

6. Please arrange back the figure for easier to capture the results.

7. Total number of crossed squares results not found in any figure.

6. PLOS authors have the option to publish the peer review history of their article (what does this mean?). If published, this will include your full peer review and any attached files.

Reviewer #1: No

Reviewer #2: **Yes: **IDRIS LONG

---

## [Author Response · Author response to Decision Letter 0]

19 Nov 2023

The response to both reviewers has been uploaded as two separate files

---

## [Editor Report · Decision Letter 1]

8 Dec 2023

A Study of Roflumilast Treatment on Functional and Structural Changes in Hippocampus in Depressed Adult Male Wistar Rats

PONE-D-23-15624R1

Dear Dr. Hassan,

We’re pleased to inform you that your manuscript has been judged scientifically suitable for publication and will be formally accepted for publication once it meets all outstanding technical requirements.

Kind regards,

Abeer El Wakil, PhD

Academic Editor

PLOS ONE

Additional Editor Comments (optional):

The concept of the present study seems interesting. The aim of the study is to investigate the impact of depression which is a common stress disability disorder affecting higher mental functions including emotion, cognition, and behavior. It may be mediated by inflammatory cytokines that interfere with neuroendocrine function, and synaptic plasticity. Therefore, reductions in inflammation might contribute to treatment response. The current study evaluates the role of Protein Kinase (PKA)- cAMP response element-binding protein (CREB)- brain derived neurotropic factor (BDNF) signaling pathway in depression and the effects of roflumilast (PDE4 inhibitor) as a potential antidepressant on the activity of the PKA-CREB-BDNF signaling pathway, histology,and pro-inflammatory cytokine production.

In my opinion, the concept of the present study is interesting, and the authors addressed the reviewers'concerns thoroughly. The manuscript in its current form is acceptable for publication at Plos One.
---

## [Editor Report · Acceptance letter]

18 Dec 2023

PONE-D-23-15624R1 

PLOS ONE

Dear Dr. Hassan, 

I'm pleased to inform you that your manuscript has been deemed suitable for publication in PLOS ONE. Congratulations! Your manuscript is now being handed over to our production team.

Kind regards, 

on behalf of

Professor Abeer El Wakil 

Academic Editor

PLOS ONE